# Fuzzy clustering for the within-season estimation of cotton phenology

**Vasileios Sitokonstantinou**[1,2☯]*, **Alkiviadis Koukos**[1☯], **Ilias Tsoumas**[1], **Nikolaos S. Bartsotas**[1], **Charalampos Kontoes**[1], **Vassilia Karathanassi**[2]

**1** National Observatory of Athens, IAASARS, BEYOND Centre of EO Research and Satellite Remote Sensing, Athens, Greece, **2** Laboratory of Remote Sensing, National Technical University of Athens, Athens, Greece

☯ These authors contributed equally to this work.
\* vsito@noa.gr

**Data Availability Statement:** Data relevant to this paper are available from Zenodo at DOI: 10.5281/zenodo.7646864 (https://doi.org/10.5281/zenodo.7646864).

## Abstract

Crop phenology is crucial information for crop yield estimation and agricultural management. Traditionally, phenology has been observed from the ground; however Earth observation, weather and soil data have been used to capture the physiological growth of crops. In this work, we propose a new approach for the within-season phenology estimation for cotton at the field level. For this, we exploit a variety of Earth observation vegetation indices (derived from Sentinel-2) and numerical simulations of atmospheric and soil parameters. Our method is unsupervised to address the ever-present problem of sparse and scarce ground truth data that makes most supervised alternatives impractical in real-world scenarios. We applied fuzzy c-means clustering to identify the principal phenological stages of cotton and then used the cluster membership weights to further predict the transitional phases between adjacent stages. In order to evaluate our models, we collected 1,285 crop growth ground observations in Orchomenos, Greece. We introduced a new collection protocol, assigning up to two phenology labels that represent the primary and secondary growth stage in the field and thus indicate when stages are transitioning. Our model was tested against a baseline model that allowed to isolate the random agreement and evaluate its true competence. The results showed that our model considerably outperforms the baseline one, which is promising considering the unsupervised nature of the approach. The limitations and the relevant future work are thoroughly discussed. The ground observations are formatted in an ready-to-use dataset and will be available at https://github.com/Agri-Hub/cotton-phenology-dataset upon publication.

## Introduction

Crop phenology is key information for crop yield estimation and agricultural management and thereby actionable knowledge for the farmer, the agricultural consultant, the insurance company and the policy maker. Crop phenology is the physiological development of the plant from sowing to harvest. The precise and timely knowledge of the growth status of crops is

**Funding:** This work has been supported by the e-shape (https://e-shape.eu/) and CALLISTO (https://callisto-h2020.eu/) projects, which have been funded from the European Union's Horizon 2020 research and innovation programme under grant agreement 820852 and 101004152, respectively. There was no additional external funding received for this study. The funders had no role in study design, data collection and analysis, decision to publish, or preparation of the manuscript.

**Competing interests:** The authors have declared that no competing interests exist.

crucial for estimating the yield early in the season, but also for taking prompt action on controlling the growth to i) maximize the production and ii) reduce the farming costs [1].

Crops' water needs are a function of the phenological stage. Using the example of cotton, which is the crop of interest in this study, there is higher water usage between the flowering and early boll opening stages than in the emergence and late boll opening stages [2]. Irrigation can be interrupted on the onset of boll opening to stop the continuous growth of cotton and allow the photosynthetic carbohydrates to start contributing to the development of bolls and not the development of leaves and flowers [3]. Therefore, we need crop phenology information in order to make irrigation recommendations towards fully utilizing the expensive and often scarce water, and at the same time reduce water stress and its potential adverse effects on the yield [4]. Irrigation is one of the many examples of how phenology can benefit the agricultural practice management. Other examples include the precise application of plant growth regulators, pest management and harvesting [1]. For instance, pix (mepiquat chloride), which is the most widely used cotton growth regulator, when applied on the early flowering stage can reduce the excessive cotton vegetation growth and therefore reduce the probability of diseases and also improve lint yield and quality [5]. In the same manner, cotton picking could be rushed prior to an anticipated extreme weather event (e.g., hail), if phenology estimations show a near complete boll opening status.

For many years, phenology has been observed from the ground, through field visits and in-situ sensors. These approaches however are expensive, time-consuming and lack spatial variability. To this end, space-borne and aerial remote sensing Vegetation Index (VI) time-series have been used to systematically monitor crop phenology over large geographic regions; often termed land surface phenology [1, 6]. The freely available Sentinel-2 data offer optical imagery of high temporal and spatial resolution that introduced new opportunities for the large-scale and within-season monitoring of phenology [7, 8].

The recently published positioning papers [1, 9, 10], identify the problem of remote phenology estimation as a fundamental one for the future of agriculture monitoring. Particularly, the authors underline the importance and the expected impact of within-season estimations at high spatial resolutions. Many of the related studies focus on the prediction of few principal phenological stages, failing to truly exploit the frequency of remote sensing data. This is mostly true because the required ground observations for training and/or evaluation are usually infrequent and lack spatial variability. This kind of ground observations are usually achieved with the use of networks of phenology stations or phenocams, which are always sparse and limited in number.

In the past two decades, there has been a number of related studies that focus on the estimation of vegetation phenology using both Earth Observation (EO) and weather data, under a wide variety of methodological frameworks. Initial approaches to the problem, many of which continue to develop to this day, offered after-season phenology estimations and were usually applied at large geographic scales using medium resolution imagery [11–15]. The term after-season indicates that phenology is estimated after the crop is harvested and thus leverages the entire data time-series. This large-scale monitoring of the dynamics of phenology has been very popular in the scientific domains of ecology and climate change monitoring [8, 11–18]. Nevertheless, detailed information that is offered within the cultivation period is very important from the perspective of the farmer. Using timely and high spatial resolution phenology predictions, farmers can protect their yield and maximize their profit. Towards this direction, there has been a number of studies that provide within-season phenology predictions at the field level [19–25]. Phenology estimation can be found in literature both as a classification and as a regression problem. For the first, the phenological cycle is divided into stages or classes that last for a given period of time [19–21]. Usually the crop growth period is broken down

into i) the sprouting, ii) the vegetative, iii) the budding, iv) the flowering and v) the ripening phases. On the other hand, phenology as a regression problem translates to predicting the day of the onset of these key phenological phases [22, 23, 25, 26].

Each crop type has its own growth cycle and hence unique characteristics with respect to i) how phenology is affected by agro-climatic conditions and ii) how responsive are the space-borne or aerial EO data with respect to the various growth stages. There are crop types for which phenology and yield are highly correlated to the vegetation canopy we see from EO platforms, e.g., tobacco. This is not the case for other crops, especially those that bear fruits or bolls, where phenology and vegetation canopy are not closely coupled. In this study, we focus on cotton (Gossypium hirsutum L.), which is a unique case with non-linear relationships between its growth and the VIs that we have in our capacity to monitor it [27–29]. In literature, one can find many publications related to the estimation of phenology for rice, barley, soybean and maize [20, 21, 26, 30, 31]. However, cotton appears to be underrepresented. When searching for phenology estimation studies on cotton, one can find only few publications that date back more than a decade [32, 33], and some more recent ones that deal with multiple crop types and do not focus explicitly on cotton [34, 35]. There are also a handful of papers that evaluate the process-based model CSM-CROPGRO-Cotton, but with small-scale experiments (few fields) [36–38]. On the other hand, there are dozens of recent papers that focus on the large-scale prediction of phenology for other major crops [39, 40]. Indicatively, there have been interesting recent studies on maize [31, 41–44], rice [43, 45, 46], wheat [43, 47, 48] and soybean [31, 41, 44].

Phenology is affected by the temperature [49, 50], the photoperiod and the effective solar radiation that enables photosynthesis [51, 52], the soil properties [53], and many other agro-meteorological parameters [54]. Indeed, in cotton phenology literature we can find older studies that use exclusively meteorological data, such as soil and/or air temperatures [33, 55, 56] and other more recent ones that combine them with optical images [21, 23, 57]. Synthetic Aperture Radar (SAR) data, usually in combination with optical images, have been mostly used for estimating rice phenology [20], but also other crop types [58]. The combination of Sentinel-2 and MODIS has been one of the most popular in the field. This is true because the Sentinel-2 missions offer data of high spatial resolution that enable information extraction at the field level, whereas MODIS data and their daily acquisitions, in contrast with the 5-days revisit period of Sentinel-2, allow for the generation of dense SITS [22, 24–26, 31, 59–61]. Other data sources found in literature include Unmanned Aerial Vehicles (UAV) [62, 63] and in-field RGB sensors [64].

There are several published studies that employ supervised Machine Learning (ML) methods for land surface phenology. For instance, the authors in [21] have used Support Vector Machines (SVM) and Random Forests (RF) to integrate field, weather and satellite data for maize phenology monitoring. Furthermore, the authors in [23, 65] have used traditional ML regressors to model plant phenology based on both satellite EO and gridded meteorological data. There have also been a few Deep Learning (DL) based approaches. One example is [59], where the authors explore the use of capsules, i.e., a group of neurons to address the issues of translation invariance prevalent in conventional Convolutional Neural Networks (CNN), to learn the characteristic features of the phenological curves. There is also a number of methods that do not use ML. A few examples of such methods include dynamic multi-temporal modeling and Kalman filtering in [24], particle filtering in [25], sigmoid modeling in [60], first-derivative analysis in [22, 58], and wavelet-based filtering and shape model fitting in [66].

In this work we exploit EO data (Sentinel-2), together with numerical simulations of atmospheric and soil parameters (i.e., soil, surface and ambient temperature, accumulated precipitation, downwards shortwave radiation and soil moisture) to address the within-season

phenology estimation for cotton at the field level. Even more, since ground truth data are scarce and expensive to collect, we predict phenological stages using clustering, in order to be truly useful in real world scenarios. We go beyond the estimation of principal phenological stages and additionally identify the fuzzy transitions between stages as individual metaclasses (two ranked labels). We focus on cotton that is a vital crop for the Greek economy and agricultural ecosystem, and even more has been underrepresented in the phenology estimation literature. Finally, we developed and made publicly available a unique dataset of cotton growth ground observations, collected by an expert who performed hundreds of field visits in Orchomenos, Greece.

## Materials and methods

### Ethics statement

The field campaigns were conducted in cotton fields in Orchomenos, Greece, which are privately owned by the members of the agricultural cooperative of Orchomenos. A memorandum of understanding was signed with the cooperative that explicitly allowed to perform visits in selected fields. During the experiments, no other specific permission was required, as only observational activities were carried out and no endangered or protected species were involved.

### Study area and field campaigns

Greece has the fourth largest production of cotton per person (approx. 29 kg) and is the number one producer in the European Union (EU), with 304,000 tons per year [67]. Cotton is extensively cultivated and is very important for the national economy, with Greece being the fifth largest exporter in the world. Unfortunately, there is no organized effort to record practice calendars and phenological observations. In Greece, cotton needs between 150 to 200 days in order to complete its phenological cycle. The duration depends on the cotton variety and the agro-climatic conditions [33].

For this study, cotton growth consists of six principal phenological stages. These refer to higher level groupings of the cotton growth micro-stages defined in the official manual for damage assessment of the Greek Agricultural Insurance Organization (ELGA) [68]. These groupings have been made after consulting experts in cotton growth and the relevant literature [69]. Fig 1 illustrates the phenology of cotton in Greece. The first stage is Root Establishment (RE), referring to the period from sowing to the development of three leaves. This stage lasts between 15 to 30 days, but this can be greatly affected by weather conditions and particularly low temperatures that can slow down the process [33]. The second stage is Leaf Development (LD) and encompasses the period from the development of the fourth leaf to the appearance of the first squares. This usually takes between 35 to 45 days, but once again it is subject to weather conditions [70]. The third growth stage is Squaring (S) that includes the period between the formation of the first squares to the appearance of the first flowers. This stage takes between 15 to 30 days. Then the first flowers open with the onset of the Flowering (F) stage that lasts for 20 to 40 days. Then follows the Boll Development (BD) stage that takes 25 to 45 days until the start of leaf discoloration and the onset of Boll Opening (BO) that lasts roughly 10 to 20 days until harvest.

It should be noted that phenology is a dynamic variable and in principle can be described in great detail, going beyond these six principal crop growth stages. At any given instance, a cotton plant can be characterized by a combination of adjacent stages. For example, during the late flowering stage, a plant would have both flowers and cotton bolls, i.e., it would be transitioning to the BD stage. One of the most popular crop growth identification scales is the

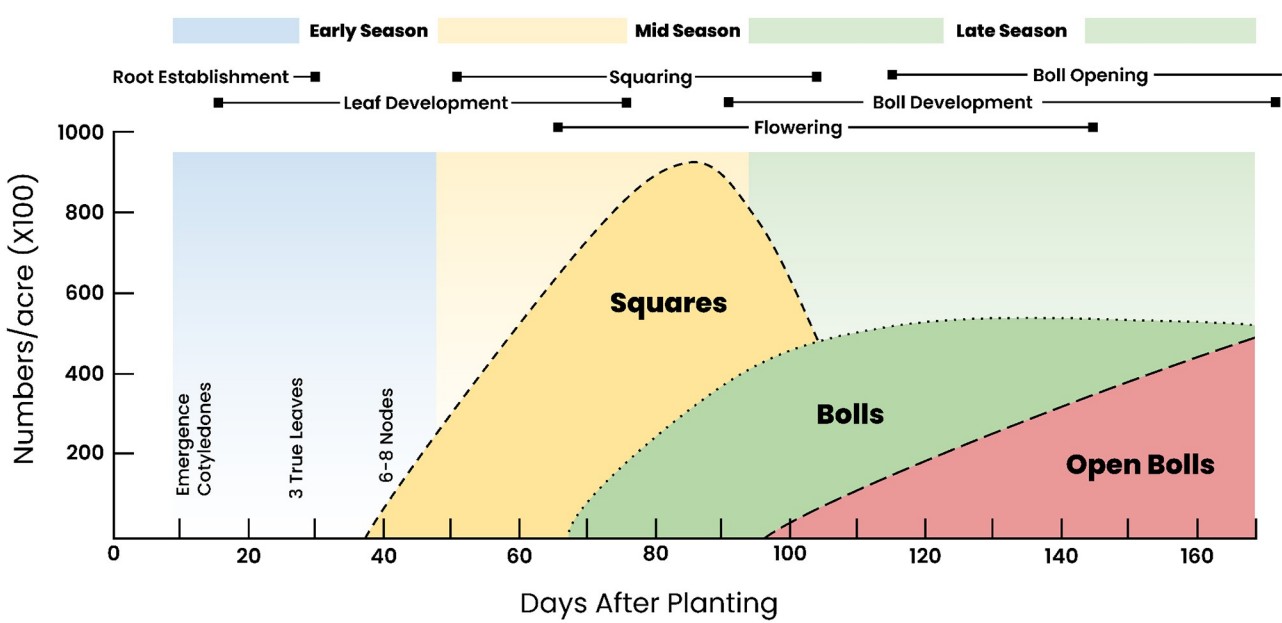

**Fig 1. The phenological cycle of cotton in Greece.** The principal phenological stages of cotton are Root Establishment (RE), Leaf Development (LD), Squaring (S), Flowering (F), Boll Development (BD) and Boll Opening (BO). The temporal overlaps between adjacent phenological stages are illustrated. Retrieved from [69] and modified.

BBCH [71]. The BBCH scale makes use of a two digit representation, with the first digit referring to the principal growth stage and the second digit describing the secondary growth stage that corresponds to an ordinal number of percentage value. The BBCH scale ranges from 00 to 99. However, collecting ground observations of this detail is a challenging task because i) a large number of samples cannot be observed in near-daily frequency and ii) it is difficult, even for experts, to assign precise growth stages, especially when this decision needs to be aggregated at the field level.

In order to collect ground truth data that would allow us to evaluate the models of this study, an agronomist, who is a cotton grower and seasoned field scouter, performed extensive and intensive field campaigns in Greece. The campaigns took place during the growing season of 2021, from root establishment to boll opening, which extends between late April and early October. The expert followed the instructions that are summarized below:

- At least 15 visits per field (approx. 3 per month) during the growing period, including at least one visit per phenological stage.

- Ideally, visit the fields in the days that Sentinel-2 passes over. If this is logistically impossible, visit the fields maximum one day prior or after the Sentinel-2 pass.

- If it is cloudy, check the next Sentinel-2 pass, consult weather forecasts and decide if the inspection could be delayed for a few days or should happen irrespective of the cloud coverage.

- Walk with a zig-zag pattern for typical scouting through the field and inspect the growth status and how it varies in space.

- Decide on the phenological stage, choosing among the six principal stages that were defined earlier, which best describes the majority of the plants in the field. If the field is in a

transitioning phase between two phenological stages, mention both and decide which is the prevailing one, i.e., the primary stage.

- Decide on the percentage that is explained by the primary and the secondary stage

- Take a panoramic photo of the entire field. Take two close-up photos of plants. The first one should be representative of the majority of the plants in the field. The second one should be representative of a minority of plants in the field. The latter close-up photo should be captured only when the percentage of the minority class, in terms of area, is deemed significant (Fig 2).

Fig 2 helps to further illuminate what is meant by the terms primary, secondary and percentage of prevalence. The close-up photo labelled "majority" shows a representative plant of this aggregate status of the the field, also conveyed by the panoramic photo, which can be described with BD as primary and BO as secondary. On the other hand, the close up photo labelled "minority" refers to a plant that is less common in the field and is more representative of the secondary stage. This becomes even more clear by inspecting the "minority" close-up photo for the 05/09 visit, which shows a plant that is well into the BO phase. The expert would identify the percentage of prevalence based on the area that the plants represented by the "majority" and "minority" close-up photos cover in the field, but also the average density of the two stages in each plant. In this example and for the 05/09 visit, BD was 100% and BO was 80% prevalent, which means that BD was found in every plant of the field while BO on 80% of the plants. Respectively, for the 17/09 visit BO was 100% and BD was 30% prevalent.

| Panoramic | Majority | Minority |
| --- | --- | --- |

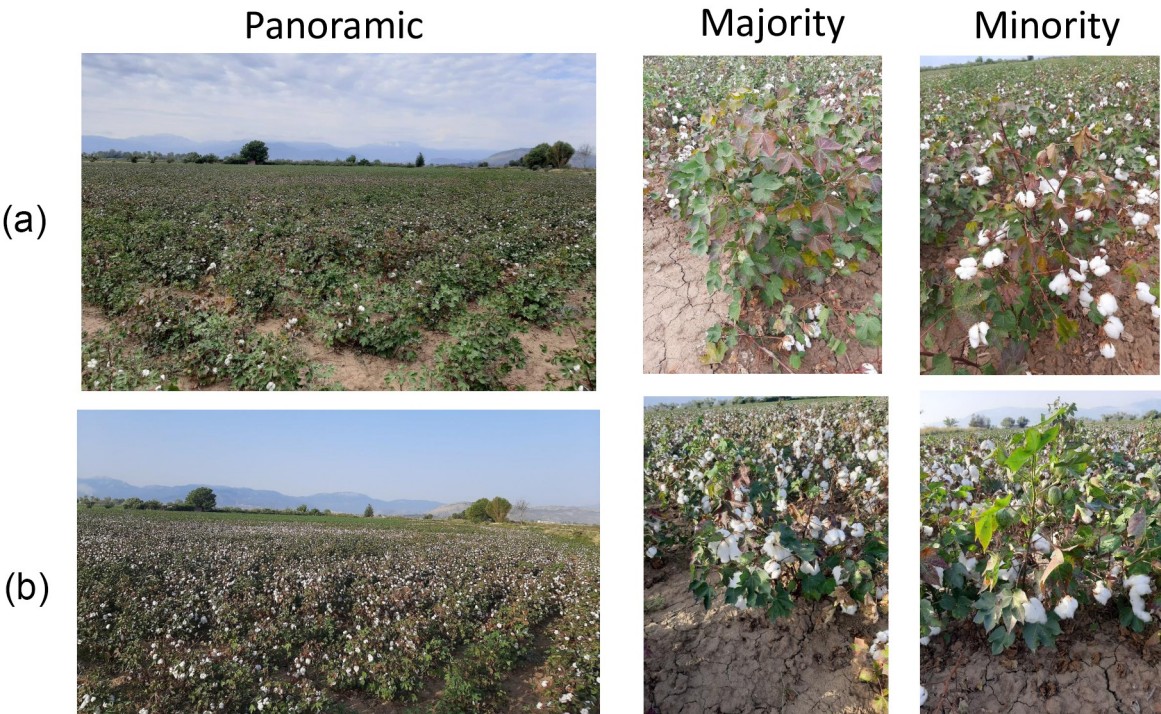

**Fig 2. Examples of field photos.** Field photos that were captured in two consecutive visits of the same field. For each visit there is a panoramic photo of the field and two close-up photos of a plant that represent the majority and minority of the field, respectively. **(a)** Visit on 05/09, when the primary stage was BD and the secondary stage was BO. **(b)** Visit on 17/09, when the primary stage was BO and secondary was BD.

**Table 1. Difference in days between ground and satellite observation pairs.**

| Difference in days | #Cloud-free S2[a] captures | Cum. Frequency(%) |
|---|---|---|
| **0** | 475 | 37 |
| **1** | 594 | 83 |
| **2** | 173 | 97 |
| **>3** | 43 | 100 |
| **Total** | 1285 | - |

[a] Sentinel-2.

During the growing season of 2021, our expert made 1285 visits to 80 cotton fields in Orchomenos. Orchomenos is an agrarian municipality in Viotia district of central Greece. The fields that participated in the ground observation campaigns are part of the agriculture cooperative of Orchomenos that has the highest selling price for cotton in Greece. It should be noted that among the 80 fields, 10 different cotton varieties were cultivated. This variability is important, as one can evaluate the performance of the phenology estimation models and draw conclusions on their generalization. The field visits were appropriately scheduled in order to have minimal differences between ground and satellite observations. In total, we acquired 67 different Sentinel-2 images, from mid March until the end of October. The mean difference between the ground and the cloud-free Sentinel-2 observations was 0.86 days and the standard deviation was 0.89 days. Table 1 depicts the distribution of the difference in days between the ground and the satellite observation pairs.

Table 2 shows the number of primary and secondary ground observations for each principal phenological stage. The expert was instructed to assign a primary stage label to any visit, thus the number of primary observations equals to the number of field visits. On the other hand, a secondary stage is not necessarily present, since it is observed only in a transitioning phase between two principal phenological stages (e.g., from flowering to boll development). Specifically, a secondary stage was observed only in 669 out of the 1285 visits (52%).

Figs 3 and 4 show the Kernel Density Estimation (KDE) of the Days of Year (DoY) for which the expert observed the different principal phenological stages as primary and secondary, respectively. It becomes clear that there are many chronological overlaps among the stages, for both the primary and secondary annotations. Inspecting Fig 3, we see that the overlaps get progressively larger as we move towards the end of the growing cycle. This is expected as

**Table 2. Distribution of phenological stages.**

| Stage | Ground observations | |
|---|---|---|
| | **Primary stage[a]** | **Secodary stage[b]** |
| **RE** | 75 | 4 |
| **LD** | 421 | 20 |
| **S** | 212 | 5 |
| **F** | 229 | 148 |
| **BD** | 252 | 315 |
| **BO** | 96 | 177 |
| **Total** | **1285** | **669** |

[a,b] The number of ground observations for each principal phenological stage of cotton that have been classified as [a]primary and [b]secondary stage labels.

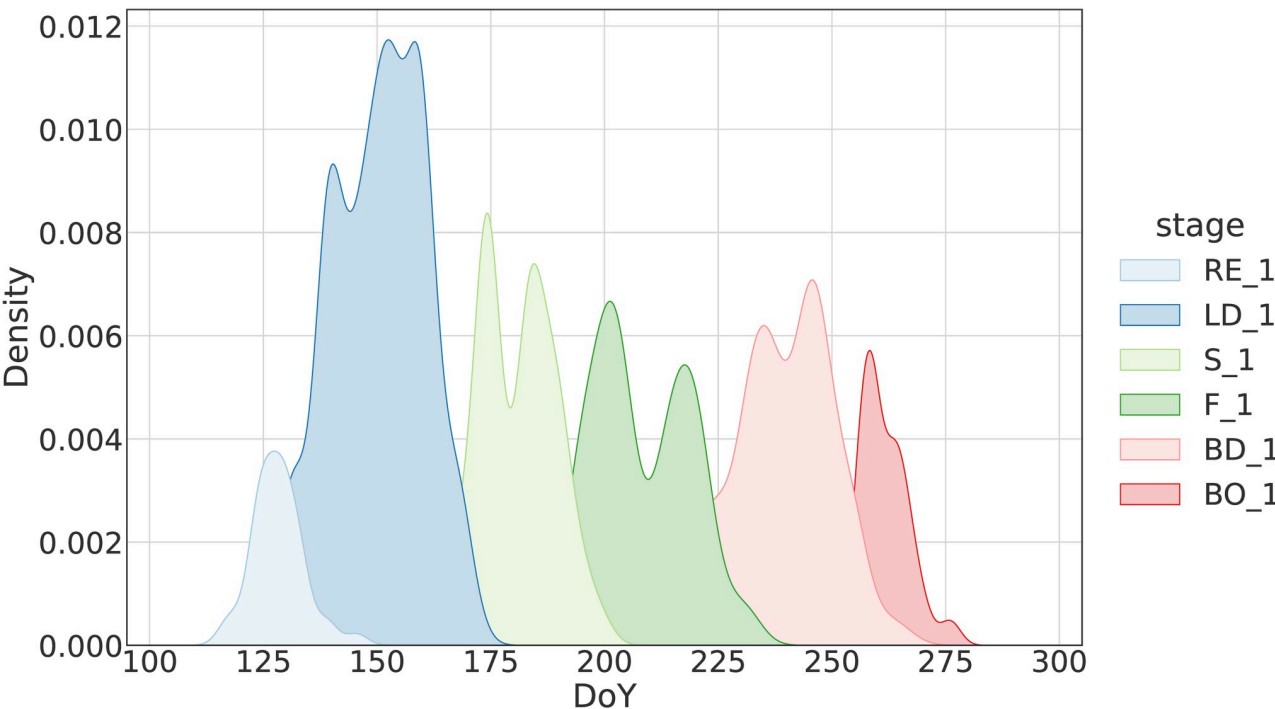

**Fig 3. Distribution of DoY for which the inspector observed the phenological stages as primary.**

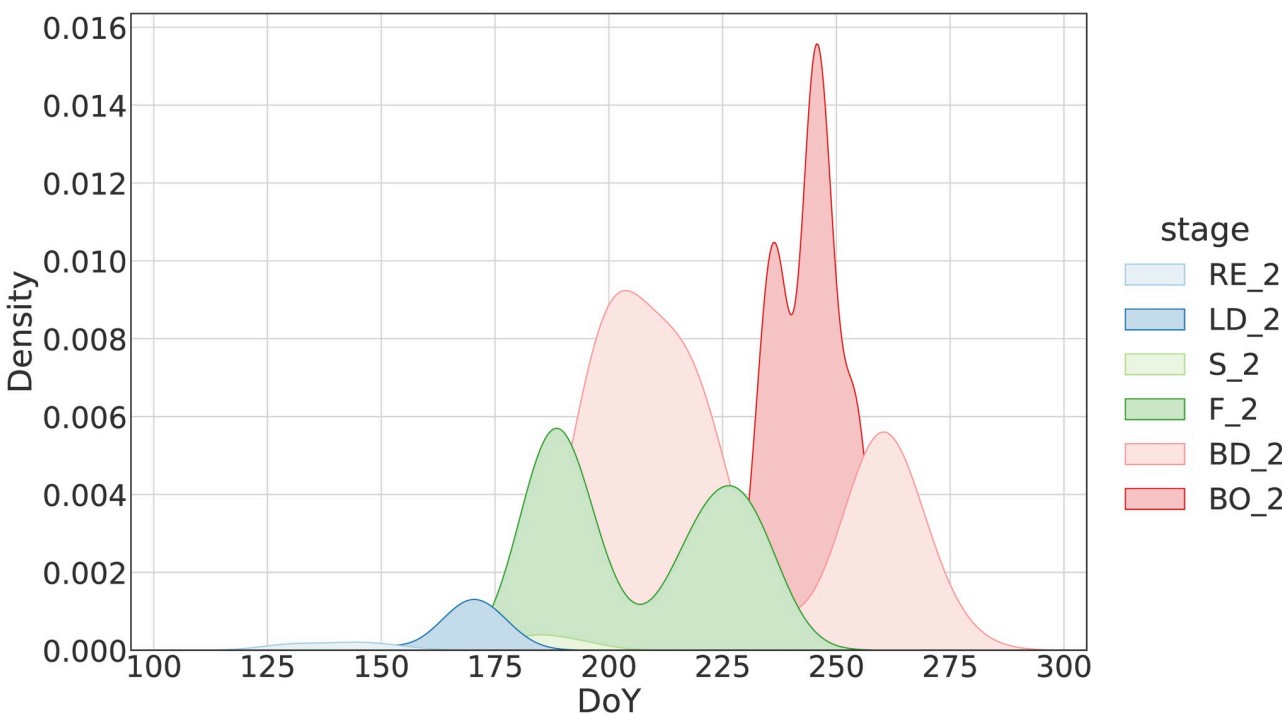

**Fig 4. Distribution of DoY for which the inspector observed the phenological stages as secondary.**

differences in growth accumulate with time and thus get more pronounced. The two figures also highlight how different the rate of growth can be even for fields that cultivate the same crop type, have similar sowing date and are in close proximity. Finally, it should be noted that for the secondary stage observations, the KDEs appear to have two modes (Fig 4). The first and second mode refer to observations that were made prior and after the onset of the "primary" phase of a phenological category. This is the reason why there are extensive overlaps among the KDEs for the secondary stage observations.

The ground observation dataset is public (https://github.com/Agri-Hub/cotton-phenology-dataset), encouraging the community to use it for training, testing and evaluating the performance of cotton phenology estimation or yield estimation models. The dataset includes a) the geographic location and geometry of fields (EPSG:4326—WGS 84), b) the days of inspection, c) the primary phenological stage and the percentage it describes, d) the secondary phenological stage and the percentage it describes, e) the sowing and harvest dates, f) a panoramic photo of the field, g) close-up photos of representative plants for the "majority" and "minority" phenological stages.

The quality of the dataset has been evaluated by another expert, Expert 2, who reviewed a randomly selected subset of 145 ground observations (11.28%) using the available panoramic and close-up photos. Expert 2 was not aware of the ground observations and was asked to decide on the primary stage and secondary stage, if there was one. Then a third expert reviewed the disagreements between the decisions of Expert 1 and Expert 2 using once more the photos captured during the visits. Table 3 shows a number of metrics for the interrater agreement.

The analysis from Expert 3 yielded the percentages of agreement and disagreement, N/A observations and undecided observations. The agreement score refers to the cases for which Expert 2 provided the same label as Expert 1. N/A observations are the observations that were considered unfair to include in the evaluation. In particular, we do not take into account 3 cases:

1. When Expert 1, i.e., the one that visited the field, provided only a primary stage observation and Expert 2 provided the same observation as a secondary stage. Such mismatch should be penalized only for the primary stage and it is considered N/A for the secondary stage.

2. When Expert 1 provided a primary stage observation with 100% prevalence and secondary stage observation with prevalence less than or equal to 60%, whereas Expert 2 gave the same primary stage observation but no secondary stage observation. Such cases should not be penalized for the secondary stage observation since the photos may not show it clearly.

3. When the primary observation of Expert 1 agrees with the secondary observation of Expert 2 and vice versa, and the prevalence percentage provided by Expert 1 is above 50%. For example, Expert 1 observes F as primary with 100% prevalence and stage BD as secondary

**Table 3. Interrater agreement metrics[a].**

|  | Primary stage | Secondary stage |
|---|---|---|
| **Agreement** | 0.72 | 0.60 |
| **Disagreement** | 0.10 | 0.09 |
| **N/A** | 0.17 | 0.20 |
| **Undecided** | 0.01 | 0.11 |
| **Krippendorff's alpha (ordinal)** | 0.95 | |

[a] For the primary and secondary growth stage annotations (Expert 1 v. Expert 2).

with 60% prevalence, whereas Expert 2 observes BD as primary and F as secondary. In fact, such cases can be very similar because both describe the transitional phase from one principal stage to the other. Based on that, and since Expert 2 judges according to a couple of photos, it is not fair to penalize these cases as wrong annotations.

The undecided category refers to instances that Expert 2 claimed and Expert 3 confirmed that the photos were not good enough to make a fair assessment. Finally, the rest of the cases are considered disagreements.

Crop growth labeling is not straightforward because one attempts to derive ordinal categories to what is actually a continuous variable. Thus, the results will be heavily dependent on the choice of the category limits, i.e., the instance a principal phenological stage transitions to the next. These limits, although pre-defined (e.g., onset of LD is the appearance of three fully formed leaves) and explained in detail to the various experts, are subject to different interpretations. The annotations collected through the field visits are ordinal and thus we need to select an appropriate measure to reveal information on the reliability. Krippendorff's alpha is a versatile interrater agreement metric that is applicable to any level of measurement, e.g., nominal, ordinal or interval [72]. The Krippendorff's alpha between Expert 2 and Experts 1 for ordinal level of measurement was 0.95. This indicates a strong agreement on the primary stage observations, which combined with the analysis performed by Expert 3, constitutes the annotation method reliable. From this point on, we only use Expert 1's ground observations. The aforementioned analysis was merely performed to assure the quality of the ground observations.

## Predictor variables

Table 4 lists the predictor variable candidates with which we experimented in this study: i) the Sentinel-2 derived products and ii) the atmospheric and soil numerical simulations. In this section we focus on the acquisition and pre-processing the various prediction variables, whereas in the next sections we elaborate on how these variables are incorporated in the feature space and feed the phenology estimation models.

**Optical images.** The optical spectrum variables used in this study were derived from Sentinel-2 images. As mentioned earlier, optical SITS have been popular in related research studies, with special attention given to Sentinel-2, but also MODIS data. The method of this work exploits only Sentinel-2 images in order to provide crop-specific phenology predictions at the field level. There are cases where the agricultural landscape is dominated by a single crop cultivation, e.g. U.S. Corn belt, and MODIS can be a tremendous help in crop-specific phenology predictions. But in Greece the landscape is fragmented and the medium spatial resolution of MODIS would yield mixed optical signatures of multiple crop types.

The optical component of this study's variable space comprises the RGB, NIR and SWIR spectral bands of Sentinel-2, but also several VIs. VIs are combinations of the spectral bands that can highlight particular vegetation properties [84]. Table 4 lists the various VIs that have been used in this work. The VIs investigated are some of the most common in the relevant literature. As the crop grows, its spectral signature changes with time. It starts with bare soil and then there are stems and leaves, and then flowers and bolls. These different phases have different biophysical and biochemical properties and thus different light reflectance profiles. Therefore we investigate multiple VIs so as to capture the maximum possible information at every stage of the growth. With regards to pre-processing, the Sentinel-2 images have been atmospherically corrected using the Sen2Cor software [85]. Additionally, clouds have been removed using the Sen2Cor scene classification product. Then, the null-valued pixels have been filled using linear interpolation on the SITS.

**Table 4. Summary of predictor variable candidates.**

| Variable | Formula[a] | Resolution[b] |
|---|:---:|:---:|
| Day of Year (DoY) | sine, cosine | - |
| Temperature at surface | min, max | 2 km |
| Growing Degree Days (GDD) 2m | $(T_{max}-T_{min})/2-T_{base}$ | 2 km |
| Accumulated Precipitation | max | 2 km |
| Downwards Shortwave Radiation | max | 2 km |
| Soil temperature 0–10 cm depth | min, max | 2 km |
| Soil moisture 0–10 cm depth | min, max | 2 km |
| Normalized Difference Vegetation Index (NDVI) [73] | (B08-B04)/(B08+B04) | 10 m |
| Normalized Difference Water Index (NDWI) [74] | (B03-B08)/(B08+B03) | 10 m |
| Normalized Difference Moisture Index (NDMI) [75] | (B08-B11)/(B08+B11) | 20 m |
| Plant Senescence Reflectance Index (PSRI) [76] | (B04-B02)/ B06 | 20 m |
| Soil-Adjusted Vegetation Index (SAVI) [77] | ((B08—B04)/(B08 + B04 + 0.428))*(1.0 + 0.428) | 10 m |
| Enhanced Vegetation Index (EVI) [78] | 2.5*(B08-B04)/((B08+6*B04–7.5*B02) + 1.0) | 10 m |
| Visible Atm. Resistant Indices Green (VARIgreen) [79] | (B03-B04)/(B03+B04-B02) | 10 m |
| Green Atmospherically Resistant Index (GARI) [80] | (B08-(B03-(B02-B04)))/(B08-(B03+(B02-B04))) | 10 m |
| Structure Insensitive Pigment Index (SIPI) [81] | (B08-B02)/(B08-B04) | 10 m |
| Wide Dynamic Range Vegetation Index (WDRVI) [82] | (0.2*B08-B04)/(0.2*B08+B04) | 10 m |
| Global Vegetation Moisture Index (GVMI) [83] | ((B08+0.1)-(B12 + 0.02))/((B08+0.1)+(B12+0.02)) | 20 m |

[a] B is the spectral reflectance value of the band number of the Sentinel-2 image.

[b] The variables with resolution of 2km are our NWP, whereas those of 10 or 20m resolution are Sentinel-2 derived products.

**Atmospheric and soil parameters.** A dense, long-term and efficient monitoring of the atmospheric state is rarely met in real-life conditions except for experimental campaigns. Automatic weather stations can contribute, but insufficient spatial coverage or bad distribution of them are typical problems, let alone temporal gaps and discontinuities due to sensor failures or human aspects (poor maintenance). Given the absence of a dense in-situ network or weather radar scans over our area of study that could provide a necessary insight, we relied upon high resolution Numerical Weather Predictions (NWP) (2 km) from our convection-permitting operational configuration of WRF-ARW model [86]. The model is initialized daily with the latest available analysis and after the exclusion of the first few hours (spin-up time) in order to reach a statistical equilibrium, the following 24-hour estimates are utilized.

While this grid spacing may appear quite coarse when compared to the resolution of the EO-derived products, we should consider that this is an outcome of NWP simulations. We are able to provide estimates of atmospheric parameters every 2 km over regions that are heavily under-monitored (in-situ weather station can be available every 100 km over croplands). This scale is considered high resolution in NWP terms and the resolving of cloud microphysical processes, such as convection, starts to happen explicitly under this spatial threshold which is particularly important to resolve fine atmospheric processes on a local scale without having to rely on parameterization schemes. A 2 km forecast fulfils our needs given that the physiographic characteristics of the crop regions we focus upon are not areas of high topographical complexity, so great gradients are not expected.

The specific parameters that were used in our pipeline were an outcome of consultation with agronomist experts and systematic literature review upon their correlation with the evolution of cotton [54]. They include Air Temperature, Surface (skin) Temperature, 0–10 cm Soil Temperature and Moisture, Precipitation and Incoming Shortwave Radiation. Growing

Degree Days (GDD) is additionally computed, as it is one of the most essential indicators of phenology. Inspecting the GDD equation in Table 4, $T_{max}$ and $T_{min}$ are the maximum and minimum daily air temperatures at 2m (from surface) and $T_{base}$ is the crop's base temperature (15.6˚C). The latter is defined as the temperature below which cotton does not develop. The GDD variable is also known as thermal time and is an indicator of the effective growing days of the crop [87].

### Fuzzy clustering

We propose a fuzzy clustering method for the within-season estimation of cotton phenology. The workflow of the proposed approach is depicted in Fig 5. We use clustering to circumvent the ever-present problem of sparse, scarce and difficult-to-acquire ground observations, which constitute the supervised alternatives of limited applicability in operational scenarios. Nevertheless, we visited tens of fields and collected hundreds of ground observations in order to test the performance of our models. The aim of our newly introduced ground observation collection protocol was to extract more information than the principal growth stages, as it is usually the case in related works. This happens at the labelling level, via taking advantage of the primary and secondary stage ground observations, as described in detail earlier.

$\mathcal{L} = \{\lambda_1, \lambda_2, \ldots, \lambda_k\}$ is the finite ordered scale of the principal cotton growth class labels, where $\lambda_1$ is RE and $\lambda_6$ is BO. The ground observation protocol allowed k = 6 phenological stages to choose from and a maximum of n = 2 labels to assign to each field, i.e., the primary

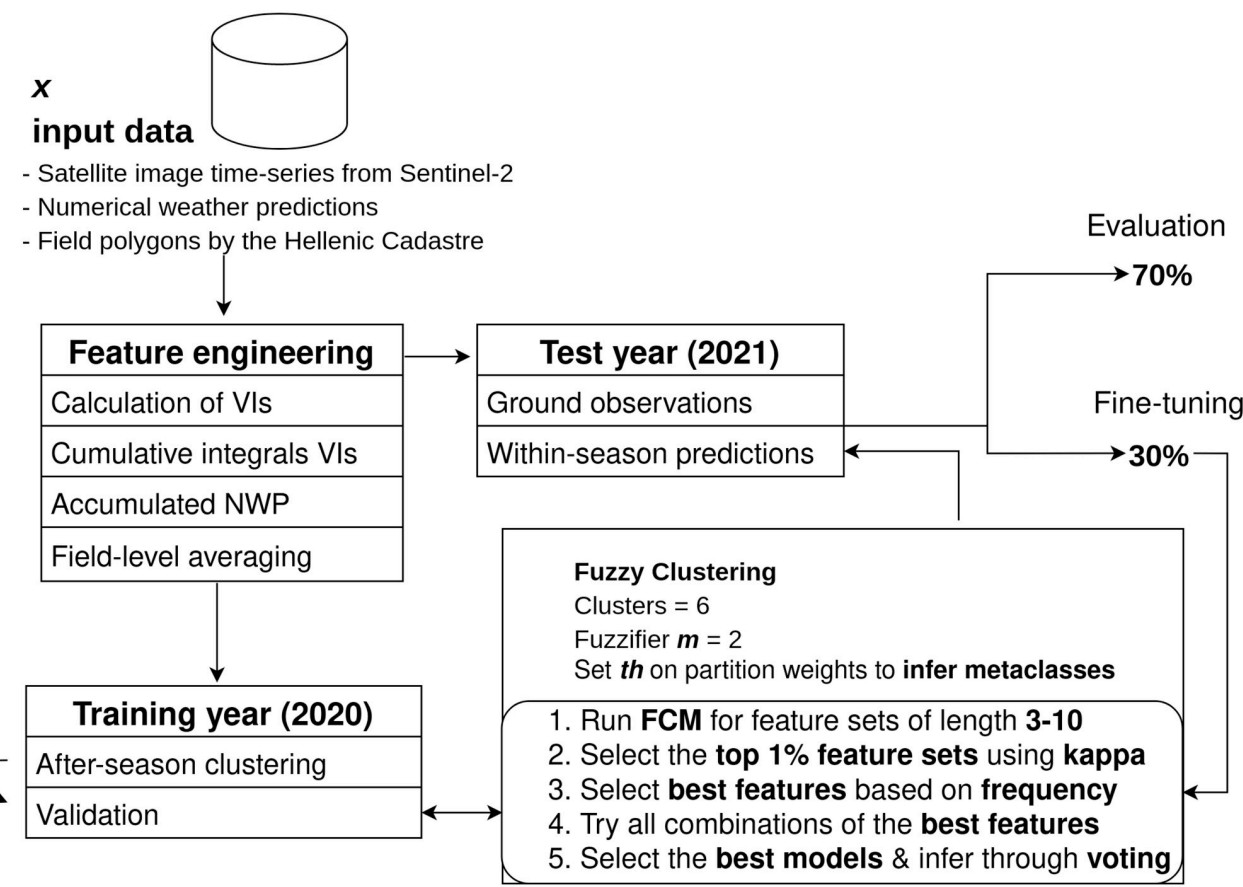

**Fig 5. The proposed methodology for cotton phenology estimation.**

and secondary stage categorization. This restricted the set of allowable labels $L_r$ to all possible permutations of n = 2 elements of $\mathcal{L}$ and all unit sets. Specifically, there are $k^2$ = 36 allowable labels. This multi-label problem can be reduced to a single-label one by considering each subset as a distinct metaclass [88]. In reality, however, there are only 16 possible metaclasses, as the primary and secondary stage for a field can differ only a single position in the ordinal scale. Eq 1 lists this set of 16 metaclasses in ordered scale.

$$L_r = \{\lambda_1, (\lambda_1, \lambda_2), (\lambda_2, \lambda_1), \lambda_2, (\lambda_2, \lambda_3), (\lambda_3, \lambda_2), \lambda_3, (\lambda_3, \lambda_4), (\lambda_4, \lambda_3), \lambda_4,$$
$$(\lambda_4, \lambda_5), (\lambda_5, \lambda_4), \lambda_5, (\lambda_5, \lambda_6), (\lambda_6, \lambda_5), \lambda_6\} \tag{1}$$

In order to estimate the phenological metaclass for each field at any given instance, we used the Fuzzy C-Means (FCM) clustering algorithm. $X \in R^{K \times E}$ denotes the element space that was used as input to the FCM, where K is equal to the number of fields (M) multiplied by the number of Sentinel-2 acquisitions (J), and E is equal to the number of features. Each row of the two-dimensional space in Eq 2 represents field $i$ at the time instance $j$ (DoY of Sentinel-2 acquisition). For the Sentinel-2 variables, each element $x_{(i, j), d}$ gets the mean value of variable $d$ for all pixels of field $i$ at the instance $j$. In the same fashion, for the NWP variables, each element $x_{(i, j), d}$ gets the value of the nearest grid cell to field $i$ for the instance $j$. To calculate the mean value of the pixels and grid cells that fall within each field we used the parcel boundaries retrieved from the Hellenic Cadastre (scale 1:5000).

$$X_{(i,j),d} = \begin{pmatrix} x_{(1,1),1} & x_{(1,1),2} & \cdots & x_{(1,1),E} \\ x_{(1,2),1} & x_{(1,2),2} & \cdots & x_{(1,2),E} \\ \vdots & \vdots & \ddots & \vdots \\ x_{(1,J),1} & x_{(1,J),2} & \cdots & x_{(1,J),E} \\ \vdots & \vdots & \ddots & \vdots \\ x_{(M,J),1} & x_{(M,J),2} & \cdots & x_{(M,J),E} \end{pmatrix} \tag{2}$$

Since phenology is dependent on the relative temporal progression of variables, we use the accumulated NWP parameters and the cumulative integrals of the VIs. The starting point for the accumulation is set around the earliest sowing DoY for the fields in the area of interest. In our case, this starting point was the 10th of April (DoY 100). Therefore, the feature space includes the Sentinel-2 VIs and their cumulative integrals, the accumulated NWP parameters, and the cosine and sine of the Sentinel-2 acquisition DoY (Table 4).

During the learning phase, the FCM algorithm attempts to partition K elements $X = \{x_1, \ldots, x_K\}$ that capture the entirety of the season into c = 6 clusters that is assumed they represent the six principal growth stages of cotton (after-season clustering in Fig 5) [89]. This is considered to be a valid assumption given that each element is described by the EO and NWP variables, their time-accumulated variants, and the associated DoY. The assumption is also supported by the results in the next section. The algorithm returns a list of $C = \{c_1, c_2, \ldots, c_6\}$ cluster centers and a partition matrix $W = (w_{k,l}) \in R^{K \times c}$, where $w_{k,l}$ is the degree to which the element $x_k$ belongs to cluster $c_l$. We applied FCM on the 2020 variable space (training year) in Orchomenos and then used the clusters C to produce within-season predictions, in dynamic fashion, during the 2021 season (test year). The training cotton fields of 2020 were 194 in total, and were extracted from a pre-trained crop classification model, based on [90].

After the clustering, the phenological stages are assigned to the different clusters via exploiting the time order. For this, the most common order of clusters is recorded and then matched to the ordered scale of labels in $\mathcal{L}$. It is common to address a multi-label problem in an indirect way using a scoring function $f : X \times \mathcal{L} \to R$ that assigns a real number to each element-label pair [88]. The assumption here is that this scoring function corresponds to the probability of each label being relevant to an element. In our case, the scoring function $f$ is the FCM and the scores are the membership grades $w_{k,j}$ of each element $x$. In other words, the FCM attempts to find the labels in $\mathcal{L}$ and then the partition scores or weights are used for multi-label prediction via thresholding. Even more, sorting the labels according to their score provides label ranking, enabling the identification of the primary and secondary stages, as given through the field inspections (Eq 3).

$$\lambda_i \leq_x \lambda_j \Leftrightarrow f(x, \lambda_i) \leq f(x, \lambda_j), i, j = 1 \ldots 6 \tag{3}$$

where $\lambda$ refers to the 6 principal phenological stages from RE to BO, $x$ is an element from the element space in Eq 2 and $f$ is the scoring function of the FCM algorithm, i.e., the partition score.

## Evaluation metrics

We considered the ML task of phenology estimation as a multi-label classification problem, given the potential duality of phenological stages at a given instance. The two labels are ranked as primary and secondary phenological stages according to their relevance, or prevalence, in the field. Therefore, the metrics for assessing our model should capture these properties.

First, we categorize the predictions to error classes according to the difference or displacement between the prediction and the ground observation in the ordinal scale of metaclasses. These error classes are labelled as diff-$o$, with $o \in \{0, 1, 2, 3\}$. For instance, if our model predicted $\lambda_2$ and the ground observation was $(\lambda_3, \lambda_2)$, then according to Eq 1 the prediction is categorized as diff-2. Similarly to the top-N accuracy, we devised the maxdiff-$o$ accuracy, with $o \in \{0, 1, 2, 3\}$, measuring the percentage of predictions with a displacement no greater than $o$. For instance, maxdiff-2 is the percentage of predictions that have at most an error of two displacement units. We also use the well known kappa coefficient, together with its linear and quadratic weighted variants. The weighted kappa metrics allow for disagreements to be weighted differently and are commonly used when labels are ordered.

We additionally incorporate the Normalized Discounted Cumulative Gain (NDCG). It is a popular metric in the world of information retrieval and specifically in tasks such as top-N ranking and item recommendations. In our case, we use the various partition weights or membership probabilities ($w_{k,j}$) as relevance values and rank the phenological clusters accordingly. We use NDCG@2, as we take into account only the top 2 ranked stages/clusters that we assume represent the primary and secondary annotations. The highly relevant phenology stage should be ranked higher than the less relevant stage, which is in turn expected to be ranked higher than non-relevant stages. NDCG@2 captures and evaluates exactly this capability of the model.

NDCG is based on the cumulative gain that simply sums the relevance scores for top@$p$ ($p = 2$). This is mathematically expressed by:

$$CG_p = \sum_{i=1}^{p} rel_i \tag{4}$$

In this case, we set $rel = 2$ for the primary stage and $rel = 1$ for the secondary stage. The cumulative gain however does not take into account the position of the phenological stage in the rank. This is done by the discounted cumulative gain, as in Eq 5, which makes use of a log-

based penalty function and reduces the relevance score that is normalized by a penalty equivalent to each position.

$$DCG_p = \sum_{i=1}^{p} \frac{rel_i}{\log_2(i+1)} = rel_1 + \sum_{i=2}^{p} \frac{rel_i}{\log_2(i+1)} \tag{5}$$

Finally, the discounted cumulative gain is simply normalized by the ideal order of the relevant items and we end up with NDCG [91].

## Results

The FCM clustering was performed on the EO and NWP variables for the Orchomenos region in 2020 (training year) and was then applied on the equivalent feature space of 2021 (test year) for the within-season prediction of phenological metaclasses. Our FCM-based approach has three parameters, i) the number of clusters, ii) the fuzzifier $m \in R$, with $m \geq 1$ and iii) the partition score threshold, above which a cluster is considered as a valid phenological stage label. The fuzzifier m was set to 2, which is commonly preferred when using the FCM algorithm [92–95]. According to [96] the best choice for m is in the interval [1.5, 2.5], with $m = 2$ being the most common choice. Finally, the partition threshold ($th_w$) depends on the distribution of the partition scores during the learning phase of the FCM algorithm.

For each element $x_k$, FCM gives a partition weight $w_{k,j}$ that refers to the degree the element belongs to each of the clusters. The weights are then sorted for each element. The two-labelled nature of our target (primary and secondary stages) implies that the values of partition weights ranked 3rd or lower, should not be considered as valid phenological stage labels. Thus, we set the threshold $th_w$ equal to the value of the 98th percentile of the partition weights ranked in the third place (Fig 6). We used the 98th percentile to eliminate the influence of potential

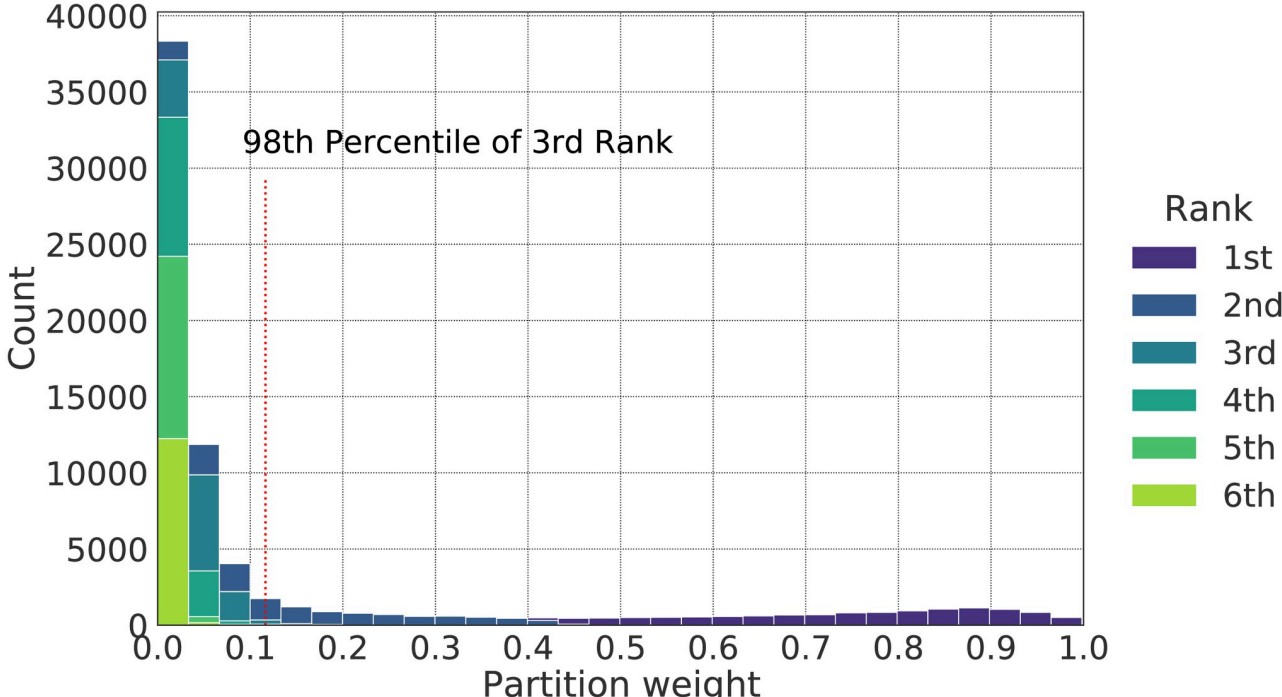

**Fig 6. Distribution of FCM partition weights for m = 2.** The hue indicates the rank in which the weights were given for each prediction. The dotted line shows the threshold above which 2nd ranked clusters are considered as valid secondary phenological stages.

outliers. Indicatively, Fig 6 illustrates the distribution of the partition weights (for $m = 2$) for a representative model. The different colors indicate the rank in which the weights were given for each prediction, i.e., from 1st to 6th. In this case, the threshold is set to 0.11 based on the aforementioned rule. A different threshold was computed for each different model that we tested.

Having configured the three parameters, we trained FCM models (after-season prediction) on the 2020 cotton fields (training year), with multiple combinations of features (~80K). The FCM algorithm works well in low dimensions. Specifically, [97] suggested that for dimensions larger than ten, the FCM starts to present ill behaviour; and for this reason we ran our experiments on hyperspaces of up to ten features. We had 32 features to choose from, i.e., the variables described in Table 4 and their cumulative variants, and we generated feature sets of length from 3 to 10 features. However, an exhaustive analysis would have required more than 200 million clusterings. To avoid running so many experiments, which is translated in months of experimentation, we fitted the FCM for feature sets of length $E \in \{3, \ldots, 10\}$ using ten thousand random feature combinations for each length. It should be noted that all feature sets include the features cos(DoY) and sin(DoY) that set the time frame.

In order to evaluate the performance of the algorithm on the different feature spaces, we applied each individual FCM model (i.e., the cluster centers C and thresholds $\text{th}_w$, from the training year) on the 2021 cotton fields, for which we have ground truth labels. We split the data into test and validation (70–30), and then used the kappa coefficient values of the validation data to find the best 1% (mean kappa = 0.45) of feature combinations. Based on these top feature combinations, we selected the 15 most appeared features. These were i) the VIs SIPI, GVMI, EVI, NDMI and SAVI, ii) the cumulative integrals of WRDVI, PSRI, NDWI and SIPI, iii) the accumulated maximum soil temperature, maximum surface temperature, maximum solar radiation and Accumulated GDD (AGDD) and iv) the always present time features of sin (DoY) and cos(DoY).

Having concluded to the aforementioned set of predictors, we ran the FCM algorithm again for the training year (2020), for every possible combination of those features in spaces of length 6 to 15. This time, apart from the kappa coefficient, we also considered the maxdiff-1 score. Specifically, based on the performance of the best 1% of feature combinations, as mentioned above, we kept solutions with kappa coefficient larger than 0.46 and maxdiff-1 larger than 0.86, which resulted to 604 cases out of 7,814. Moreover, by setting these values as such, we ensure that we acquire better solutions compared to a baseline model. The baseline model refers to an FCM with only DoYs as input. Since phenology is closely related to the DoY, the baseline is used to capture this chance agreement and showcase by comparison the real competence of our model. Detailed comparisons with the baseline model follow later in this section.

The analysis revealed that for most cases feature sets of length $E > 10$ yielded sub-optimal results. Specifically, from the top 604 models, 84% contained no more than 10 features. Besides the DoY features, the AGDD is by far the most common feature, since it appeared in more than 86% of the best solutions. Another important observation here is that the cumulative integrals of VIs and the accumulated NWP features are more important than than the single-date VIs. The importance of these features is great since they also capture the dimension of time, which is essential for an unsupervised phenology prediction model. Nevertheless, the results showed that the majority of well-performing feature sets contained at least one feature from each category, namely VIs, cumulative integrals of VIs and accumulated NWPs.

From the top 15 features, the cumulative integral of SIPI, the accumulated maximum solar radiation, NDMI and SIPI did not appear as frequently in the best 604 models. For this reason, we discarded models that included these features. The final set of models, which was used for our predictions, comprised models with 8 and 9 features that included at least one feature

**Table 5. Metrics of performance for our phenology prediction model and the baseline model.**

| | Ours | Baseline |
|---|---|---|
| **maxdiff-0** | 0.53 | 0.38 |
| **maxdiff-1** | 0.88 | 0.86 |
| **maxdiff-2** | 1.00 | 0.97 |
| **maxdiff-3** | 1.00 | 1.00 |
| **Cohen's kappa** | 0.48 | 0.33 |
| **Weighted kappa (Linear)** | 0.88 | 0.84 |
| **Weighted kappa (Quadratic)** | 0.98 | 0.97 |
| **NDCG** | 0.93 | 0.88 |

from each category. This resulted to a total of 82 models. In order to ensure the generalization and robustness of our methodology, by not depending on a single feature set, we generate the final predictions through majority voting on those best models. The 82 feature sets are listed in S1 Table.

Table 5 shows the performance of our model and the baseline model on the test set. Indeed, it is shown that our model provides a significantly larger number of diff-0 predictions, namely perfect agreements between predicted and ground truth metaclasses. This is also evident via observing Cohen's kappa that is notably higher for our model. In terms of absolute values though, the model shows moderate performance in these two metrics. However, given the unsupervised nature of the FCM algorithm as well as the fact that it works very well in the other metrics and avoids outlier errors, we claim that the overall performance is satisfactory and the proposed approach shows potential. It is also worth mentioning how our model significantly outperforms the baseline in terms of NDCG, denoting a better ranking capacity.

Table 6 shows the prediction errors in metaclass displacement units, for phenology metaclasses that had at least 10 ground observations. The displacement units are computed via multiplying the normalized confusion matrix with a weight matrix, for which the cells one off the diagonal of the confusion matrix are weighted 1, those two off are weighted 2, etc. Then the weighted displacement units are summed for each ground observation metaclass. Our model offers a smaller average displacement for six out of the eight metaclasses that account for the majority of ground observations.

It is observed that the metaclass (BO, BD) offers the smallest error in displacement units, whereas the (F, BD) metaclass gives by far the largest. This can be explained by the fact that

**Table 6. Prediction errors in metaclass displacement units.**

| | Metaclass | Support | Displacement | |
|---|---|---|---|---|
| | | | **Ours** | **Baseline** |
| 1 | (RE, -) | 72 | **0.41** | 1.33 |
| 4 | (LD, -) | 415 | **0.62** | 0.89 |
| 6 | (S, LD) | 17 | **0.41** | 1.00 |
| 7 | (S, -) | 122 | 0.58 | **0.17** |
| 8 | (S, F) | 73 | **0.30** | 0.56 |
| 11 | (F, BD) | 225 | 1.04 | **0.92** |
| 12 | (BD, F) | 75 | **0.37** | 0.75 |
| 14 | (BD, BO) | 177 | **0.54** | 0.72 |
| 15 | (BO, BD) | 90 | **0.03** | 0.68 |
| | **Average** | 1266 | **0.48** | 0.78 |

**Table 7. Confusion matrix for the six principal phenological stages.**

| | | Pred | | | | | |
|---|---|---|---|---|---|---|---|
| | | **RE** | **LD** | **S** | **F** | **BD** | **BO** |
| Truth | **RE** | 71 | 4 | 0 | 0 | 0 | 0 |
| | **LD** | 39 | 361 | 21 | 0 | 0 | 0 |
| | **S** | 0 | 6 | 195 | 11 | 0 | 0 |
| | **F** | 0 | 0 | 12 | 204 | 13 | 0 |
| | **BD** | 0 | 0 | 0 | 28 | 195 | 29 |
| | **BO** | 0 | 0 | 0 | 0 | 3 | 93 |

that metaclass (BO, BD) is at the edge of the growing cycle and can only be confused with the metaclass that precedes it. On the other hand, metaclass (F, BD) is at the vegetation peak, where plants are well into the flowering phase and some bolls have started to develop. The consecutive metaclasses (F, -), (F, BD) and (BD, F) are situated near the plateau that is formed around the peak of the VI time-series curve (or valley, given the VI). Therefore, there are not significant differences in VI values among the three metaclasses, which explains the less than optimal performance for metaclass (F, BD). Overall, the average displacement shows significant difference between the two models. Our model achieves a respectable average error of less than half a metaclass.

Table 7 shows the confusion matrix for the hard clustering predictions. It can be observed that for the principal phenological stages the model performs rather well, with an overall accuracy 87%. Most misclassifications are observed for the BD stage. This is actually expected given the number of observations for which BD is observed in one of the transitional metaclasses (Table 6). As a matter of fact, BD is never observed as unit set metaclass.

As mentioned, the expert would visit each field three or four times a month. Thus, it was common to observe a field in a particular crop growth stage for multiple consecutive visits (3 to 6). The chronological order of observation, i.e., the relative position of the ground observation in the range enclosed between the first and last time a stage was observed as primary for a particular field, is useful since it indicates if it is observed in its early, middle or late phases.

Fig 7 shows the distribution in the order of observation for the different disagreement categories, diff-*o*. The order of observation is categorized into early, middle and late visits. For a field that was observed in a particular phenological stage for three to five consecutive visits, early visit was the first visit and late visit was the last visit. In the seldom cases that a phenological stage was observed in six consecutive visits, then the first two and last two would be characterized as early and late visits, respectively.

Fig 7 shows that the majority of the perfect agreements, diff-0, were mostly middle and late observations. On the other hand the vast majority of big disagreements, i.e., diff-1, diff-2 and diff-3, were for early stage observations or, for fewer cases, late stage observations. This is expected, as middle observations would indicate that the field is well into a particular stage, whereas early or late observations denote transitional phases.

## Discussion

### Crop phenology

The results indicated that using clustering for the within-season phenology estimation is promising but challenging. EO and NWP data are competent predictor variables for this type of problems, as they represent both the land cover changes and the crop growth drivers, but cannot fully capture the physiological growth stages of crops. Furthermore, for operational

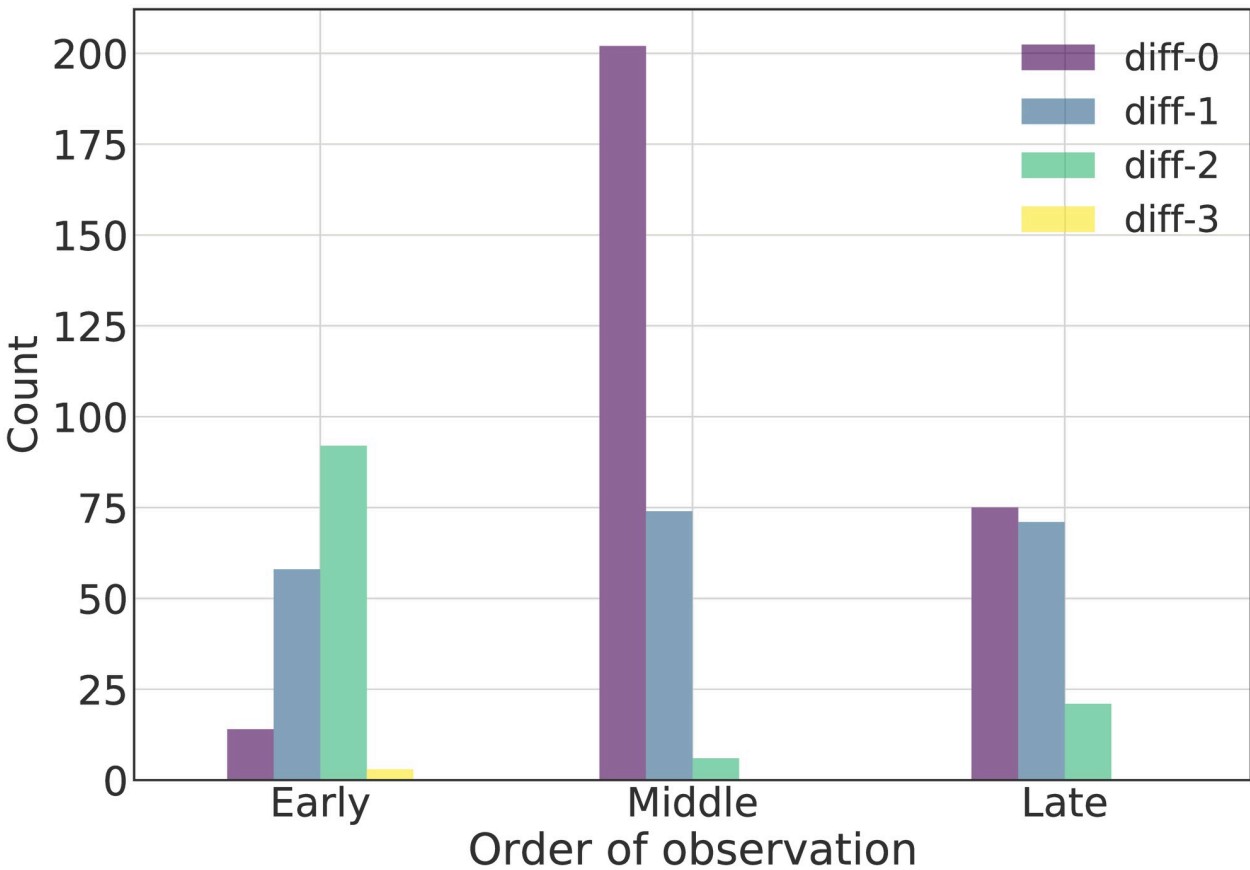

**Fig 7. Distribution of the chronological order of observation.** Early describes ground observations that are found at the beginning of consecutive observations of a stage, with regards to chronological order. Late, similarly, describes ground observations that are found at the end of consecutive observations of a stage and Middle contains ground observations that are found in middle positions The hues show the predictions according to their diff-*o* categorization of prediction error.

applications in agricultural management, phenology predictions should be made at a within-season basis. However, there are still several challenges in this undertaking. First, within-season predictions are possible using only a part of the data time-series. Additionally, temporally dense EO time-series are crucial to ensure that critical phenological changes are detected as soon as possible [1]. The density of the time-series however is subject to certain trade-offs. Considering the freely available satellite images, higher temporal resolution implies lower spatial resolution. Furthermore, optical SITS is significantly affected by cloud coverage. This might have not proved a problem for this study, having its area of interest in Greece, but is certainly an issue for other parts of the world. Finally, crop growth stages might not be directly related to EO and NWP atmospheric and soil variables. For this, there is a clear gap in the literature for sophisticated modelling that would be able to capture these complex relationships.

## Ground observations

Ground observations are required to assess how well the EO and NWP based land surface phenology relates with the actual phenological stages. Most validation datasets are focused only on few stages using aggregated statistics over large areas [1]. The National Agriculture Statistics Service (NASS) crop progress reports is such an example. Field-level ground observations are limited and rarely systematic. Recently, phenocams have been used to evaluate land surface

phenology approaches [98]. However, the network of phenocams is still sparse and confined. Furthermore, labeling phenological stages is a complex process and cannot be fully solved by observing photos from the field. Therefore, ground observations are still a necessity.

In this regard, this study offers a unique dataset of ground observations at the field level. The ground observations are accompanied by a panoramic and a couple of close-up photos of representative plants. We introduce a new protocol for collecting ground observations for crop growth that allows up to two label assignments. If two labels are assigned then the inspector should specify which one is the primary growth stage and which one is the secondary growth stage that describes the field. This allows for the detailed description of crop growth stages through the metaclasses that result from combining the primary and secondary stage annotations. Additionally, not having to decide on a single label makes the ground observation easier and can potentially increase the number of people who can perform them. This is true since the choice of the limits that define the principal growth stages differs among studies and ground observation protocols. Even more, closer to the start and end of those limits it gets tricky to decide on a single label. Having two labels can enable the reliable and large-scale crowdsourcing of ground observations.

Furthermore, the reliability of the ground observation collection method has been thoroughly evaluated. For this, we used the blinded interpretation of the field photos by an expert. Then the decisions of the field inspector and the photo inspector were evaluated by a third expert that decided on the percentage of agreement between the two. The quality assurance process yielded satisfactory results, deeming the ground observations reliable. The community is thus encouraged to use the openly available dataset and test their own models. The dataset is accompanied by the photos captured during the visits, which can be used for further interpretation but also computer vision tasks, such as crop classification and phenology classification.

## Clustering for phenology estimation

It was shown that the introduced clustering method managed to learn from the complete time-series of 2020 and successfully infer the phenological metaclasses in a within-season fashion for 2021. Our model significantly outperforms the baseline, making the proposed approach very promising. Furthermore, the model predicts 16 different metaclasses and goes beyond the 6 principal phenological stages, extracting more information on their transitions. This is particularly important since more intricate and precise agricultural management is now possible at the field level.

In many studies, phenology estimation is addressed as a regression problem, aiming to predict the DoYs of growth stage onsets, which is essential information for operational applications. This study's metaclass approach can also be viewed as a classification-based alternative of onset detection. The fuzzy metaclasses $(\lambda_a, \lambda_b)$ and $(\lambda_b, \lambda_a)$ denote this transitional phase between principal growth stages. In other words, the end of stage $a$ and the onset of stage $b$, respectively. In future work, further testing will be conducted for the evaluation of the spatial and temporal generalization of the proposed methodology. This will require additional ground observations at different areas and years of inspection.

We showed that formulating phenology estimation as a clustering problem, via incorporating the time in our features, is valid. The authors suggest that there is great potential and encourage the community to test more models on the proposed premise. During experiments, it was observed that the FCM was sensitive on the DoY of the first and last element included in the learning phase. This is expected since the clustering is largely dependent on time component of our features. It is therefore important to set the "start" and "end" instances with the aim to enclose the average length of the cotton season. This is easy when the sowing and

harvest dates are known, but could also be approximated via observing the mean and standard deviation of the VI time-series.

This work relied on i) feature engineering, incorporating the time in the form of cumulative EO and NWP variables, and ii) feature selection to decide on a set of optimal feature spaces. Tens of thousands of experiments were performed for feature selection, yielding robust results. The robustness lies in the fact that the top 15 features systematically appeared in the best performing models. Furthermore, phenology predictions are based on the majority vote of the best-performing combinations of the top features, making sure our approach can generalize.

Having said that, there is a number of recent studies that look into DL based unsupervised change detection on SITS [99–102]. We see great potential in such approaches and we believe could be applicable in the proposed unsupervised premise for phenology estimation. Common denominator of these methods is the learning of a smaller latent or embedding space, in which entities that bear resemblance are located closer to each other. This is particularly important for clustering techniques that aim to group similar samples in the hyperspace. Usually, clustering algorithms, such as FCM, measure this similarity among entities using pair-wise distances. It is known that high dimensional spaces are not ideal for distance based techniques, as they usually fail to capture meaningful clusters. In addition, a latent manifold representation is not greatly dependent on feature engineering and can generalize well.

## Conclusion

In this paper we proposed a fuzzy clustering method for the within-season phenology estimation for cotton in Greece. Our method is unsupervised to tackle the problem of sparse, scarce and hard to acquire ground observations. It provides predictions within-season and thus enables its usage in operational agricultural management scenarios. It focuses on cotton, which is important for three reasons—i) it is an underrepresented crop type in the related literature, ii) the relationship between remote sensing phenology and the physiological growth of cotton is complex and iii) cotton is a very important crop for the economy and agricultural ecosystem of Greece, which is the study area.

We conducted field visits to collect ground observations that are offered to the community as a ready-to-use label dataset. For this, we used a new protocol that leverages two ranked labels. This makes the observations easier and at the same time provides enhanced information on the growth status. Therefore, we approach the problem as a multi-label one, introducing the notion of metaclasses. We go beyond the principal phenological stages of cotton by providing prediction for 16 metaclasses, using the membership probabilities of the FCM classifier.

Finally, we experimented with numerous combinations of features, including accumulated numerical simulations of atmospheric and soil paramaters, Sentinel-2 based VIs and their cumulative integral variants. Based on these experiments, we provided a list of optimal feature sets that can be used for cotton phenology estimation through majority voting.

## Supporting information

**S1 Table. The feature sets of the top 82 FCM models with size 8 or 9 features.** With (I) we show the cumulative integrals of the VIs. max_soil refers to the cumulative maximum soil temperature, max_surf to the cumulative maximum surface temperature and wkappa to the linear weighted kappa coefficient. The last five columns refer to the performance metrics.
(PDF)

## Acknowledgments

The authors express their gratitude to Mr. Vaggelis Dedes for performing the ground observation campaign and Dr. Dimitra A. Loka, Researcher at the Institute of Industrial and Forage Crops in Greece, for her valuable consultations. Finally, the authors acknowledge the farmers' association of Orchomenos (ASOO) for letting us perform the ground observations on their fields and for giving consent to release these in a publicly available dataset.

## Author Contributions

**Conceptualization:** Vasileios Sitokonstantinou, Alkiviadis Koukos.

**Data curation:** Vasileios Sitokonstantinou, Alkiviadis Koukos, Ilias Tsoumas, Nikolaos S. Bartsotas.

**Formal analysis:** Vasileios Sitokonstantinou, Alkiviadis Koukos.

**Investigation:** Vasileios Sitokonstantinou, Alkiviadis Koukos.

**Methodology:** Vasileios Sitokonstantinou, Alkiviadis Koukos.

**Supervision:** Charalampos Kontoes, Vassilia Karathanassi.

**Visualization:** Vasileios Sitokonstantinou, Alkiviadis Koukos, Ilias Tsoumas.

**Writing – original draft:** Vasileios Sitokonstantinou.

**Writing – review & editing:** Vasileios Sitokonstantinou, Alkiviadis Koukos, Ilias Tsoumas, Charalampos Kontoes, Vassilia Karathanassi.

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
