## [Decision Letter · Decision Letter 0]

28 Jun 2022

PONE-D-22-14150Fuzzy clustering for the within-season estimation of cotton phenologyPLOS ONE

Dear Dr. Sitokonstantinou,

Thank you for submitting your manuscript to PLOS ONE. After careful consideration, we feel that it has merit but does not fully meet PLOS ONE’s publication criteria as it currently stands. Therefore, we invite you to submit a revised version of the manuscript that addresses the points raised during the review process.

We look forward to receiving your revised manuscript.

Kind regards,

Chi-Hua Chen, Ph.D.

Academic Editor

PLOS ONE

Journal Requirements:

Reviewers' comments:

Reviewer's Responses to Questions

**Comments to the Author**

1. Is the manuscript technically sound, and do the data support the conclusions?

Reviewer #1: Yes

2. Has the statistical analysis been performed appropriately and rigorously? 

Reviewer #1: No

3. Have the authors made all data underlying the findings in their manuscript fully available?

Reviewer #1: Yes

4. Is the manuscript presented in an intelligible fashion and written in standard English?

Reviewer #1: No

5. Review Comments to the Author

Reviewer #1: In this paper, the authors propose an approach for the within-season phenology estimation for cotton at the field level, considering a variety of Earth observation vegetation indices derived from Sentinel-2 and numerical simulations of atmospheric and soil parameters.

The paper is interesting and clear.

The authors developed and made publicly available an original dataset of cotton growth ground observations.

The Introduction is well written. The problem and the objective of the study are clearly presented.

Some comments:

The abstract need slight adjustments.

All the acronyms should be defined (e.g., UAV, ML, CNN, etc.).

A graphical representation of cotton phenological stages could help.

The number of satellite images used and the statistics associated could be given in a table (also)

I have several doubts related to agreement metrics (L226-249).

Table 3 - a new column with the origin of the data could be added.

The limitations of using NWP data at 2 km of spatial resolution should be discussed.

The results presented in Table 4 should be properly discussed.

Fig 6 could be replaced by a table.

In several parts of the manuscript, the authors need to be more specific.

In general, a language revision is recommended.

All my comments are given in the PDF file attached.

6. PLOS authors have the option to publish the peer review history of their article (what does this mean?). If published, this will include your full peer review and any attached files.

Reviewer #1: No

---

## [Author Response · Author response to Decision Letter 0]

4 Aug 2022

Dear Editor, Reviewer, 

we thank you very much for your valuable comments and critiques, which have improved the clarity of our manuscript. Accordingly, we have thoroughly revised our manuscript based on your comments and suggestions. The changes in the relevant manuscript, are highlighted in blue for added words, or red and strikethrough for deleted text. Additionally, minor language, grammatical and stylistic errors have been corrected. Our specific point-by-point responses to the comments and queries have been addressed and itemized as follows: 

Journal Requirements (Editor’s comments): 

 Please ensure that your manuscript meets PLOS ONE's style requirements, including those for file naming. The PLOS ONE style templates can be found at https://journals.plos.org/plosone/s/file?id=wjVg/PLOSOne_formatting_sample_main_body.pdf and https://journals.plos.org/plosone/s/file?id=ba62/PLOSOne_formatting_sample_title_authors_affiliations.pdf. 

The manuscript is formatted using Plos One LATEX style formatting and we have checked again and ensured that it meets all style requirements. 

 Please note that PLOS ONE has specific guidelines on code sharing for submissions in which author-generated code underpins the findings in the manuscript. In these cases, all author-generated code must be made available without restrictions upon publication of the work. Please review our guidelines at https://journals.plos.org/plosone/s/materials-and-software-sharing#loc-sharing-code and ensure that your code is shared in a way that follows best practice and facilitates reproducibility and reuse. 

We thank the editor for this comment. In this work we have not used any author-generated code. We only use open-source Python libraries, such as skfuzzy and scikit-learn, to implement the proposed methodology and run our experiments. 

 We note that you have stated that you will provide repository information for your data at acceptance. Should your manuscript be accepted for publication, we will hold it until you provide the relevant accession numbers or DOIs necessary to access your data. If you wish to make changes to your Data Availability statement, please describe these changes in your cover letter and we will update your Data Availability statement to reflect the information you provide. 

We thank the editor for this important comment. Our data will be available according to PLOS ONE requirements, after the final acceptance for publication. 

 Please include your full ethics statement in the ‘Methods’ section of your manuscript file. In your statement, please include the full name of the IRB or ethics committee who approved or waived your study, as well as whether or not you obtained informed written or verbal consent. If consent was waived for your study, please include this information in your statement as well. 

We thank the editor for this valuable comment. We have added the ethics statement in the manuscript as follows: 

“The field observations were conducted in cotton fields in Orchomenos, Greece, which are privately owned by the members of the agricultural cooperative of Orchomenos. A memorandum of understanding was signed with the cooperative that explicitly allowed to perform visits in selected fields. During the experiments, no other specific permission was required, as only observational activities were carried out and no endangered or protected species were involved.” 

Moreover, we should mention that we have also considered all the necessary actions to make this data publicly available to the community. 

We thank the editor for this helpful comment. We have not cited any papers that have been retracted. However, in the revised manuscript we have included new citations according to the reviewer’s comments and suggestions, which are explicitly mentioned in our detailed answers below. 

Reviewer's comments: 

 The abstract need slight adjustments. 

Adjustments have been made in the abstract according to the reviewer’s suggestions. 

 All the acronyms should be defined (e.g., UAV, ML, CNN, etc.). 

All acronyms have been defined in the revised manuscript. 

 A graphical representation of cotton phenological stages could help. 

We would like to thank the reviewer for this comment. A graphical representation of cotton phenological stages has been added - Figure 1 (revised document). 

 The number of satellite images used and the statistics associated could be given in a table (also) 

The newly added Table 1 in the revised document presents the difference in days between the field visit (ground observation) and the satellite image acquisition. Also, in L214 we mention the number of satellite images that were used. 

 I have several doubts related to agreement metrics (L226-249). 

We would like to thank the reviewer for this comment. Indeed, after carefully reading this section, we agree that this can be confusing. We have revised this paragraph (L255-L279 in the revised manuscript) as well as Table 3 with the interrater metrics. 

The analysis presented in this section was done to evaluate the annotations of the expert (Expert 1) that visited the fields. Since the expert took photos of the field for each visit, we used those photos to ensure that there are not any systematic mistakes. Another expert (post-doc in plant science with specialization in cotton) annotated what she sees in the field photos, and then a third expert (PhD in machine learning for sustainable agriculture) compared the results of the two other experts. From the comparison we did not find any systematic errors. There is a percentage of disagreement around 10% (depending on the stage) which can be attributed to i) the fact that Expert 2 assigns labels only by looking at 2-3 photos (cannot be compared with being in the field) and ii) the ever-present subjectivity in such annotations that cannot be truly avoided. We hope the new description of the metrics and what they represent will clarify things. Again, this section is merely used for the sanity check that our phenology annotation method does not suffer from systematic errors. 

 Table 3 - a new column with the origin of the data could be added. 

 Because the table is wide, we added this information in the footnote caption. 

 The limitations of using NWP data at 2 km of spatial resolution should be discussed. 

We would like to thank the reviewer for the comment. While in comparison to the scales of EO-derived products the aforementioned grid spacing may appear quite coarse, we should consider that this is an outcome of Numerical Weather Prediction simulations. Through an implementation of a state-of-the-art NWP model (WRF-ARW v.4) we are able to provide estimates of atmospheric parameters every 2 km over regions that are heavily under-monitored (in-situ weather station can be available every 100 km over croplands in such regions that are not urban). This scale is considered high-resolution in NWP terms and the resolving of cloud microphysical processes, such as convection, starts to happen explicitly under this spatial threshold which is particularly important to resolve fine atmospheric processes on a local scale without having to rely on parameterisation schemes. 

In cases of specific test areas and complex terrain we have delved into sub-km scales in the past [Bartsotas et al, 2016], but 2-km as an operational forecast for the whole country fully fulfils our needs given that the physiographic characteristics of the crop regions we focus upon are not areas of high topographical complexity so great gradients are not expected. Finally, the in-situ measurements from the members of farmer’s cooperatives that we deliver our products are steadily in par with the forecasted parameters (anecdotal). This explanation has been added in L336-346. 

Bartsotas, N.S., E. Nikolopoulos, E. Anagnostou, S. Solomos, and G. Kallos, 2016: Moving toward Sub-kilometer Modeling Grid Spacings: Impacts on Atmospheric and Hydrological Simulations of Extreme Flash Flood–Inducing Storms. J. Hydrometeor. doi:10.1175/JHM-D-16-0092.1. 

 The results presented in Table 4 should be properly discussed. 

We would like to thank the reviewer for this comment. Further explanation on the results has been added in L529-532 (revised document). Moreover, we should mention that the results and the limitations of our methods are also discussed in the Discussion section. 

 Fig 6 could be replaced by a table. 

Table 7 (revised document) has replaced Figure 6. 

 In several parts of the manuscript, the authors need to be more specific. 

We have addressed each comment on the pdf individually and performed a thorough revisit of the entire manuscript with many refinements that can be seen in the pdf with tracked changes. 

 In general, a language revision is recommended. 

Language revisions have been performed throughout the revised manuscript (see the tracked changes). 

Detailed comments on pdf: 

 L19: could be better discussed 

We would like to thank the reviewer for the comment. We have added an explanation in the revised manuscript L20-25 

 L25: see also: Duarte, et al. (2018). PhenoMetrics: An open source software application to assess vegetation phenology metrics. Computers and Electronics in Agriculture, 148, 82–94 

We would like to thank the reviewer for the suggestion. This reference has been added in L30 of the revised manuscript. 

 L44: give exemples (references) 

References have been added in the corresponding sentence in L49 of the revised manuscript. 

 L53-55: could be better described 

We would like to thank the reviewer for the comment. We have included a new description in the revised manuscript (L60-61). 

 L66: a reference is need to support this sentence 

We would like to thank the reviewer for the comment. We have included references to support this claim in L72 in the revised manuscript. 

 L69: give examples & L71: combine what? please identify the meterorological data what do they refer to: 

 We have revised the manuscript in L87-97 by adding examples and explanations. 

 L93: Specify those parameters 

 We have specified these parameters in L111-112 of the revised manuscript. 

 L94: ground truth data are scarce and expensive to collect (but is there data available?): 

There are only few crop phenology datasets at the parcel level, and specifically for cotton we could not find any. 

 L99: “significantly underrepresented in the phenology estimation literature” improve this justification: 

We would like to thank the reviewer for this comment. We have improved this justification in L72-82, which the first time we state that cotton is underrepresented in literature. 

 L105: can you give numbers? 

The information has been added in L131-132 (revised manuscript) 

 L115: any report? 

We would like to thank the reviewer for this comment. These groupings were defined based on consultations with agronomists with specialization in cotton, having [1] as a reference (that we have now added in L142 revised manuscript). 

[1] Oosterhuis DM. Growth and development of a cotton plant. Nitrogen nutrition of cotton: Practical issues. 1990; p. 1–24 

 L116-127: a graphical representation of cotton phenological stages could help. 

We would like to thank the reviewer for this comment. We have added Figure 1 (revised manuscript) of a graphical representation of cotton phenological stages. 

 L142: What kind of expert? 

An agronomist with Msc in smart farming and precision agriculture, and a cotton grower, which is now mentioned in L169-170 in the revised manuscript. 

 L165: should be better explained: 

We would like to thank the reviewer for this comment. The meaning of this sentence is that the expert could only use the six predefined phenological classes as labels, namely RE, LD, S, F, BD, BO. This sentence has been removed since it is stated earlier in the manuscript. 

 Fig 1: put (a), (b), ... in each photo and change the figure caption according 

This comment has been addressed in Figure 2 in the revised manuscript. 

 L189-190: this information could be given in an table (also the number of satellite images used) 

We would like to thank the reviewer for this comment. We have included a new table (Table 1 revised manuscript) with this information and we have added the number of satellite images in L214 of the revised manuscript. 

 L195: clarify 

We have added a brief clarification on this in L222-223 of the revised manuscript. 

 L198: expert? 

Yes. We have changed it. 

 Table1: put as a footnote 

Done 

 L214: coordinate system? EPSG code? 

We have added this information in L243 of the revised manuscript. 

 L220: give the % 

We have added this information in L249 of the revised manuscript. 

 L233: but the verification was carried out on the field or by the photos? 

Yes, it has been stated before that both Expert 2 and 3 decided on the primary and secondary stages for each observation only by looking at the available photos. Expert 1 was the only one who had provided labels from physical field inspections. Specifically, in L248-253 of the revised manuscript it is mentioned “Expert 2, who reviewed a randomly selected subset of 145 ground observations (11.28%) using the available panoramic and close-up photos. Expert 2 was not aware of the ground observations and was asked to decide on the primary stage and secondary stage, if there was one. Then a third expert reviewed the disagreements between the decisions of Expert 1 and Expert 2 using once more the photos captured during the visits.” 

 L239: confused... 

We would like to thank the reviewer for this valuable comment. We have revised this section completely (L255-279 revised manuscript) 

 L252: more details are needed 

We would like to thank the reviewer for this comment. Explanation has been added in L282-283 of the revised manuscript 

 L256: a reference is needed 

A reference has been added in L289 of the revised manuscript. 

 L272: removed 

 L277-282: repetitive 

We believe that reviewer was referring to the previous sentence i.e. “In some studies these two sources of optical images are used together, leveraging the high spatial resolution of Sentinel-2 data, and the high temporal resolution and thermal infrared bands of the MODIS sensor” in L275-277, which has now been removed in the revised manuscript (see L302-310) 

 L285: several 

Changed 

 L287: the original reference for each index could also be introduced in the table 

We would like to thank the reviewer for this comment. References for all vegetation indices have been added to Table 4 (revised manuscript) 

 L300: or bad distribution 

We would like to thank the reviewer for this suggestion. We have included this in L327 of the revised manuscript 

 L304: this is not a huge limitation 

This has been answered in detail in the main comments 

 L307: ??? 

The numerical model spin-up necessity is a well-known necessity in the numerical modeling community to reach a statistical equilibrium. We attach a recent study discussing this aspect in detail [2]. In a nutshell, it has to do with the initialization of regional NWP models at fine scales, that takes place through the interpolation of global estimates from coarser forecasting models. During the first few hours of the simulation, convective and wind artefacts can be evident, that dissolve as numerical resolving reaches a stable state. Hence the first few hours [6-12] of a simulation are not considered the most accurate. In our case we do not use the first 12h and keep hours 12h-36h from each run to represent the best part of each estimate (not too far in the future to diverge, not too early to have spin-up issues). 

[2] Short, C.J. & Petch, J. (2022) Reducing the spin-up of a regional NWP system without data assimilation. Quarterly Journal of the Royal Meteorological Society, 148(745), 1623–1643. Available from: https://doi.org/10.1002/ qj.4268 

 L310: references are needed 

We have added reference in L349 of the revised manuscript. 

 L314: From what? 

Air temperature at 2 meters above the surface 

 L337: following was removed 

 L348: scale 

We have added this information in L388 of the revised manuscript. 

 Eq.3: all equation parameters should be listed 

We have listed all parameters in Eq.3 in L420-422 of the revised manuscript 

 L415: repetetive 

Removed 

 L419: Can be applied here?: 

Yes, it can be applied here too. As we mention in L458-459 the interval [1.5, 2.5] is proposed as the best choice, while m=2 is the most common choice among a plethora of publications. For sanity check, we run some preliminary experiments to monitor the effect of m to the result, and indeed we conclude that m=2 is a suitable value for this problem too. 

 L433: quantify 

This is explained in the next sentences. 

 L440: Why ten?: 

Based both on the literature and on some preliminary results we run, we observed that low dimension feature spaces performed well and adding more features to a satisfactory solution did not enhance further the accuracy, on the contrary sometimes it provided worse results. Moreover, we should highlight that many features (either NWP or VIs) are highly correlated. Thus, our initial aim is to find the correlated features that are most suitable ones to solve the problem. 

The first experimentation took place to identify those features that when used produce satisfactory predictions. Unfortunately, we could not run every possible combination of features since this could result in months of experimentation. However, by selecting a fixed random subset of combinations, we give each feature the same probability to be included in a feature set and therefore to highlight its importance. For example, both the accumulated maximum precipitation and the accumulated maximum soil temperature features appeared in ~3.3% of all the feature sets. However, for the first one, from those 3.3% feature sets only 0.6% produced satisfactory results. On the other hand, for the second feature this percent in 13.5%. Therefore, as it is explained in L485-495 (revised manuscript), we conclude in a set of 15 most dominant features and for those we run every combination. 

All the remaining comments have been answered by the answers that we provided to the main comments of the reviewer. 

 L458: ? 

The best 1% solutions that were selected in the experiments of the previous paragraph produced a mean kappa score of 0.45. Thus, we set the minimum threshold of the metrics to be above that value. This explanation has been also added in the revised manuscript in L499-503

---

## [Decision Letter · Decision Letter 1]

14 Feb 2023

Fuzzy clustering for the within-season estimation of cotton phenology

PONE-D-22-14150R1

Dear Dr. Sitokonstantinou,

We’re pleased to inform you that your manuscript has been judged scientifically suitable for publication and will be formally accepted for publication once it meets all outstanding technical requirements.

Kind regards,

Muhammad Tayyab Sohail

Academic Editor

PLOS ONE

Additional Editor Comments (optional):

Reviewers' comments:

Reviewer's Responses to Questions

**Comments to the Author**

1. If the authors have adequately addressed your comments raised in a previous round of review and you feel that this manuscript is now acceptable for publication, you may indicate that here to bypass the “Comments to the Author” section, enter your conflict of interest statement in the “Confidential to Editor” section, and submit your "Accept" recommendation.

Reviewer #1: All comments have been addressed

2. Is the manuscript technically sound, and do the data support the conclusions?

Reviewer #1: Yes

3. Has the statistical analysis been performed appropriately and rigorously? 

Reviewer #1: Yes

4. Have the authors made all data underlying the findings in their manuscript fully available?

Reviewer #1: Yes

5. Is the manuscript presented in an intelligible fashion and written in standard English?

Reviewer #1: Yes

6. Review Comments to the Author

Reviewer #1: As I stated in my first revision, this work is interesting and clear. The Introduction is well-written, and the objective of the study is clearly presented. Regarding this revision, the authors made a huge effort to answer to all the reviewer comments. The results were clearly presented, the discussion was improved and in general, the language was also improved.

7. PLOS authors have the option to publish the peer review history of their article (what does this mean?). If published, this will include your full peer review and any attached files.

Reviewer #1: No

---

## [Editor Report · Acceptance letter]

27 Feb 2023

PONE-D-22-14150R1 

Fuzzy clustering for the within-season estimation of cotton phenology 

Dear Dr. Sitokonstantinou:

I'm pleased to inform you that your manuscript has been deemed suitable for publication in PLOS ONE. Congratulations! Your manuscript is now with our production department. 

Kind regards, 

on behalf of

Dr. Muhammad Tayyab Sohail 

Academic Editor

PLOS ONE